# Two Polysaccharides from Liupao Tea Exert Beneficial Effects in Simulated Digestion and Fermentation Model In Vitro

**DOI:** 10.3390/foods11192958

**Published:** 2022-09-21

**Authors:** Siqi Qiu, Li Huang, Ning Xia, Jianwen Teng, Baoyao Wei, Xiaoshan Lin, Muhammad Rafiullah Khan

**Affiliations:** 1School of Light Industry and Food Engineering, Guangxi University, Nanning 530004, China; 2Department of Food Engineering, Pak-Austria Fachhochschule, Institute of Applied Sciences and Technology, Mang, Haripur 22620, Pakistan

**Keywords:** Liupao tea polysaccharides, simulated digestion, bioactivities, in vitro fermentation

## Abstract

Liupao tea is an important dark tea, but few studies on purified Liupao tea polysaccharide (TPS) are reported in the literature. In this study, two TPSs, named TPS2 and TPS5, with molecular weights of 70.5 and 133.9 kDa, respectively, were purified from Liupao tea. TPS2 contained total sugar content (53.73% ± 1.55%) and uronic acid content (35.18% ± 0.96%), while TPS5 was made up of total sugar (51.71% ± 1.1%), uronic acid (40.95% ± 3.12%), polyphenols (0.43% ± 0.03%), and proteins (0.11% ± 0.07%). TPS2 and TPS5 were composed of Man, Rha, GlcA, Glc, Gal, and Ara in the molar ratios of 0.12:0.69:0.20:0.088:1.60:0.37 and 0.090:0.36:0.42:0.07:1.10:0.16, respectively. The effects of TPS2 and TPS5 on digestion and regulation of gut microbiota in hyperlipidemic rats were compared. In simulated digestion, TPS5 was degraded and had good antioxidant effect, whereas TPS2 was not affected. The bile acids binding capacities of TPS2 and TPS5 were 42.79% ± 1.56% and 33.78% ± 0.45%, respectively. During in vitro fermentation, TPS2 could more effectively reduce pH, promote the production of acetic acid and propionic acid, and reduce the ratio of *Firmicutes* to *Bacteroidetes*. TPS5 could more effectively promote the production of butyric acid and increase the abundance of genus *Bacteroides*. Results indicate that polysaccharides without polyphenols and proteins have better antidigestibility and bile acid binding. Meanwhile, polysaccharides with polyphenols and proteins have a better antioxidant property. Both have different effects on the gut microbiota.

## 1. Introduction

Different types of teas such as green, white, yellow, oolong, black, and dark tea are available. Similar to the famous Puerh and Fuzhuan teas, Liupao (Liubao) tea is one of the typical representatives of dark tea. Liupao tea is mainly produced in Wuzhou, Guangxi Zhuang Autonomous Region, and is one of the Chinese national geographical indication products. In 2021, Wuzhou produced 25,000 tons of Liupao tea with a comprehensive output value of 11 billion yuan (RMB). “Wuzhou Liupao Tea” was named as the most powerful tea brand in China in 2021 [1]. Liupao tea is made with *Camellia sinensis* var. *sinensis*, fermented at 15–18% water content and 40–55 °C for 15–30 days, followed by aging for three years [2]. During fermentation, the distinctive flavor of Liupao tea is formed by the reaction of moisture and heat in the presence of *Aspergillus* and *Staphylococcus* microorganisms, among others. Fermentation decreases the content of thearubigin, total phenols, and total flavonoids, and increases theabrownine and soluble carbohydrate [2]. The alteration of components may form a distinctive color with mellow taste and generate a woody flavor of Liupao tea. Fermentation degrades the outer wall of Liupao tea with the help of microorganisms, which may be associated with the enhancement of polysaccharide (TPS) in tea [3].

TPS is an active substance and its physicochemical properties can be defined by components, weight, monosaccharide composition, configuration of glycosidic linkage, and the position or site of the glycosidic linkage. Different raw materials or purification processes could lead to different properties of dark TPS. The molecular weight of crude Fuzhuan TPS is 748–889 kDa, comprising galacturonic acid, galactose, and arabinose, containing 31.8–41.6 mg GAE/100 g polyphenols, 1.3–5.0% protein, 31.8–41.6% uronic acid, and 45.2–59.4% total sugar [4]. The molecular weight of crude Puerh TPS is 631–3900 kDa, mainly composed of galactose, arabinose, and mannose, containing 4.23–19.96% protein, 32.63–40.66% uronic acid, and 14.36–20.20% total sugar [5]. Thus, crude dark TPS is mainly composed of galactose, mannose, and arabinose; it has a large molecular weight; and contains a certain amount of protein and polyphenols. The purification of the crude TPS was conducted by chromatographical techniques. The purified Fuzhuan TPS obtained mostly comprised galacturonic acid and galactose, containing 40.4% uronic acid and 44.78% total sugar, and having a molecular weight of 741 kDa [6]. The purified Puerh TPS was mainly composed of arabinose, galactose, and glucose, containing 7.16% protein, 12.10% uronic acid, and 52.79% total sugar, having a molecular weight of 251.2 kDa [7]. These findings indicated that the polyphenol and protein contents of crude dark TPS were reduced, the molecular weight was more concentrated, and galactose was the main component after purification.

TPSs have many bioactivities, such as antioxidant, hypoglycemic, hypolipidemic, antiaging, and anticoagulant [8]. The activity of crude TPS is generally accepted to be superior to the purified TPS due to the higher protein and polyphenol contents in crude TPS. However, it has been found that the activity of TPS is not associated with polyphenols and proteins, but associated with uronic acid, molecular weight, and monosaccharide composition [9]. For example, the antioxidant capacity of Qingzhuan TPS was attributed to polyphenols, while the free radical scavenging activity of green TPS was linked with the carboxyl group in hexanoic acid rather than polyphenol compounds [5]. Fermenting Oolong tea increased the content of protein. Thus, the antioxidation of deeply fermented Oolong tea was enhanced. However, paradoxically, the free radical scavenging capability of unfermented green TPS was better when compared with the fully fermented black TPS [9]. Therefore, the relationship between the physicochemical properties and activity of dark TPS is still not completely clear. Liupao tea is a typical post-fermentation tea. Because of different geographical locations, processing technologies, and production materials, the composition of Liupao TPS is different from that of other TPSs. Therefore, studying the effect of Liupao TPS properties on its activity is necessary.

A previous study has shown that the crude Liupao TPS is a heteropolysaccharide containing polyphenols and proteins, and it had a protective effect on hyperlipidemic rats [10]. Moreover, fermentation can reduce the molecular weight of the polysaccharides in Liupao tea, change the polysaccharide structure, and improve the ability of anticoagulation and bile acid binding [11]. However, these studies were based on in vivo study of crude Liupao TPS and in vitro study of purified Liupao TPS. Thus, whether the purified Liupao TPS has bioactivity in the digestive tract deserves further exploration. In this paper, two different purified TPSs were isolated from Liupao tea, and the dynamic changes in the chemical properties and antioxidant and bile acid binding capacities during simulated digestion in vitro were studied. Furthermore, the effects on the gut microbial population and the production of short-chain fatty acids (SCFAs) in hyperlipidemic rats were analyzed. The results provide a theoretical reference for the development of Liupao TPS as a health food and have important implications for the promotion of traditional health food development.

## 2. Materials and Methods

### 2.1. Materials and Chemicals

The Liupao tea sample (three grade tea) was provided by Wuzhou Zhongcha Tea Factory Co., Ltd. (Wuzhou, Guangxi, China).

Sephadex G-100 gel, serial standard monosaccharides (glucose (Glc), glucuronic acid (GlcA), xylose (Xyl), galactose (Gal), rhamnose (Rha), mannose (Man), and arabinose (Ara)), 1-phenyl-3-methyl-5-pyrazolone (PMP), bovine serum albumin (BSA), 1,1-diphenyl-2-picryl-hydrazil (DPPH), 3-ethylbenzothiazoline-6-sulfonic acid (ABTS), different molecular-weight dextran standards, and pancreatin were obtained from Sigma-Aldrich (St. Louis, MO, USA). Polyamide resin, DEAE-52 cellulose, and Sephadex G-100 gel were purchased from Solarbio (Beijing, China). Cholestyramine, glycocholic acid, glycodeoxycholic acid, glycodesoxycholic acid, taurochenocholic acid, taurodeoxycholic acid, heparin, and inulin were supplied by Yuanye Bio-Technology Co., Ltd. (Shanghai, China). Bile acid, total cholesterol (TC), triglyceride (TG), and low-density lipoprotein cholesterol (LDL-C) analysis commercial kits, were provided by Nanjing Jiancheng Bioengineering Institute (Nanjing, China). The standard products of SCFAs (acetic acid, propionic acid, and butyric acid) were acquired from Zhenzhun Biotechnology Co., Ltd. (Shanghai, China). All reagents were of analytical grade and obtained from Shanghai Chemistry and Reagents Co., Ltd. (Shanghai, China).

### 2.2. Preparation and Purification of TPS2 and TPS5

TPS was obtained following the method of Ying et al. [12] with slight modification. Briefly, Liupao tea was treated with 80% ethanol at 1:15 (*w*/*v*) ratio for 24 h to remove the theabrownin, oligosaccharides, and other small molecular substances. After filtration, the residue was extracted twice with 70 °C water at the ratio of 1:15. The extract was then concentrated by an RE-3000A rotary evaporator (Yarong Biochemical Instrument Factory, Shanghai, China) at 65 °C and 60 rpm and then precipitated by using 95% ethanol at 4 °C for 12 h. After centrifugation, the sample was dissolved in water and lyophilized. Thus, crude TPS was obtained that was further dissolved in water at 70 °C; pigments and protein were removed using polyamide resin to obtain refined Liupao TPS.

The refined Liupao TPS concentrate was lyophilized and fractionated by using a DEAE–52 cellulose column (φ2.6 × 50 cm). Four polysaccharide fractions were obtained, namely, TPS0, TPS1, TPS2, and TPS5 (elutied with distilled water and 0.1, 0.2, and 0.5 M NaCl solution, respectively). TPS2 and TPS5 were further passed through a Sephadex G–100 gel chromatograph at a flow rate of 0.3 mL/min to collect more concentrated molecular weight fractions. Lastly, the eluents were freeze dried by using an Alpha 2–4 LD and vacuum freeze dryer (CHRIST, Osterode, Germany) and measured by the phenol–sulfuric acid method.

### 2.3. In Vitro Simulated Saliva–Gastrointestinal Digestion of TPS2 and TPS5

In vitro simulated saliva digestion and gastrointestinal digestion were carried out following the methods of Huang et al. [13] and Chen et al. [14], respectively. The simulated salivary juice (1.0 L) was composed of NaCl (0.76 g), KCl (1.49 g), CaCl_2_ (0.13 g), and α-amylase (431.25 mg). The concentration of polysaccharide solution was 2.0 mg/mL. The experiment was designed as follows: group A consisted of polysaccharide solution and simulated salivary juice at equal concentrations, while group B consisted of simulated salivary juice and distilled water at equal concentrations. Both groups were incubated in a water bath at 37 °C for 0.5 h and then immediately immersed in boiling water for 5 min to inactivate the amylase for further analysis.

Gastric electrolyte solution was made by dispersing 3.1 g of NaCl, 1.1 g of KCl, 0.15 g of CaCl_2_, and 0.6 g of NaHCO_3_ in 1.0 L distilled water, and the pH was adjusted to 3 with 0.1 M of HCl. The simulated gastric juice was composed of 37.5 mg of gastric lipase, 35.4 mg of pepsin, 1.5 mL of CH_3_COONa solution (1 M, pH 5.0), and 150 mL of gastric electrolyte. The final pH of the gastric juice was adjusted to 3 with 0.1 M HCl. In brief, 50 mL of sample (2.0 mg/mL) or ultrapure water was blended with 50 mL of simulated gastric juice, and then digested in a water bath at 37 °C. At 0, 2, 4, and 6 h, 5 mL of solution was taken out and immediately dipped in boiling water for 5 min to inactivate the enzyme for further analysis.

Intestinal electrolyte solution was prepared by dispersing 540 mg of NaCl, 65 mg of KCl, and 33 mg of CaCl_2_ in 0.1 L distilled water, and the pH was equilibrated to 7 with 0.1 M NaOH. Then, 6.5 mg of parenzyme, 200 g of the bile salt solution (4%, *w*/*w*), and 50 g of the pancreatin solution (7%, *w*/*w*) were added to 50 g of the simulated intestinal juice, and the pH was adjusted to 7.5 with 0.1 M of NaOH to afford the simulated small intestinal juice. The reaction solution after 6 h of simulated gastric digestion was adjusted to a pH of 7–7.5 with 1 M of NaHCO_3_. Then, the reaction solution was blended with the simulated small intestinal juice at a ratio of 10:3. The mixture was digested with a shaker (37 °C, 200 r/min). At 0, 2, 4, and 6 h, 5 mL of solution was taken out and immediately dipped in boiling water for 5 min to inactivate the enzyme for further analysis.

### 2.4. Chemical Characterization of TPS2 and TPS5 and Its Digested Fractions

#### 2.4.1. Compositional Analysis

The phenol–sulfuric acid method with Glc as a standard was used to measure the total sugar content [15]. Protein was determined according to Bradford’s method, using BSA [16]. Uronic acid was determined by the phenyl–carbazole method using glucuronic acid [17]. Total polyphenol content was quantified by Folin–Ciocalteu assay reagent, using gallic acid [18]. Reducing sugar content was determined by the 3,5–dinitrosalicylic acid method [19].

Monosaccharide composition and free monosaccharide were measured by high-performance liquid chromatography (HPLC) following the method of Chen et al. [20]. The polysaccharide sample (400 μL of 2 mg/mL) was mixed with 400 μL NaOH solution (0.3 M) and 400 μL methanol solution of PMP (0.5 M). Then, the reaction mixture was heated at 70 °C for 100 min. Next, 500 μL of HCl solution (0.3 M) was added to neutralize the resultant solution. Then, 5 mL of trichloromethane was added and the solution was mixed completely to remove the excess PMP and then passed through a 0.45 μm membrane. The monosaccharide compositions and free monosaccharide of Liupao TPS and its digestive production were determined on an HPLC equipped with a photodiode-array detector (e2695, Waters, Milford, MA, USA). Detection was set at 250 nm and the column (C18, 4.6 mm × 150 mm, 5 μm) temperature was 30 °C; a 10 µL sample was injected. The mobile phase consisted of ammonium acetate solution (0.02 M) and acetonitrile at the ratio of 83:17 (*v*/*v*), with a flow rate of 1.0 mL/min. As a reference, the same conditions were used for the standard.

#### 2.4.2. Molecular Weight Distribution

Molecular weight was determined by high-performance gel permeation chromatography (HPGPC) on an Agilent 1260 Infinity system (Agilent, Palo Alto, CA, USA) installed with a refractive index detector and a Shodex OHpak SB-804HQ column (8.0 mm × 300 mm, Showa Denko, Tokyo, Japan). The sample solution (2 mg/mL) was filtered (0.22 µm) and injected into the column. The mobile phase was a 0.02% NaN_3_ solution with a flow rate of 0.5 mL/min, and the column temperature was 35 °C. Linear regression was calibrated with a dextran series standard (5, 12, 50, 410, and 670 kDa).

#### 2.4.3. Fourier Transform Infrared (FTIR) Spectra

FTIR spectroscopy (VERTEX 70, Bruker, Ettlingen, Germany) was used to analyze the molecular structure of TPS2 and TPS5 with KBr tablet method using 400–4000 cm^−1^ wavelength.

#### 2.4.4. Ultraviolet-Visible Spectra

Ultraviolet-visible spectra of the polysaccharide samples (2 mg/mL) were characterized using UV-1601 PC spectrophotometer (Shimadzu, Kyoto, Japan) at a scanning range of 200–400 nm.

### 2.5. Antioxidant Activities of TPS2 and TPS5 and Its Digested Fractions

The ABTS radical scavenging and DPPH were determined by the method of Wu et al. [18].

### 2.6. Bile Acid Binding In Vitro

Bile acid binding ability was determined by the method of Qin et al. [11] with slight modifications. A stock solution (36 mM) was prepared by mixing glycocholic acid (9 mM), glycochenocholic acid (9 mM), glycodeoxycholic acid (9 mM), taurochenocholic acid (4.5 mM), taurodeoxycholic acid (4.5 mM), and phosphate buffer solution (pH 6.3, 0.1 M) and stored at −20 °C in a freezer. Before utilizations, the solution was diluted with phosphate buffer to 0.72 µmol/mL concentration. Then, 1 mL of the sample (2 mg/mL) or 50 mg of cholestyramine was mixed with 1 mL HCl (0.01 M) and incubated at 37 °C for 1 h. Afterward, the pH was adjusted to 7.3 with 0.2 M of NaOH. The working solution and 4 mL of pancreatin (10 mg/mL) were added to the reaction mixture and incubated for 2 h at 37 °C. After centrifugation, the supernatant was frozen at −20 °C and bile acids were measured calorimetrically using commercial kits (Nanjing Jiancheng Bioengineering Institute, Nanjing, China). The results were calculated by the following formula:Bile acid binding capacity (%) = [(A_mixture_ − A_supernatant_)/A_mixture_] × 100(1)
where A_mixture_ is the bile acid concentration in the positive blank and A_supernatant_ is the bile acid concentration in the supernatant.

### 2.7. In Vitro Fermentation

#### 2.7.1. Animals and Diets

All procedures were conducted in accordance with the “Guiding Principles in the Care and Use of Animals” (China) and were approved by the Animal Experiment Ethics Committee of Guangxi University (GXU2018-066). Fifteen healthy Sprague–Dawley rats, 6 weeks old, weighing 200 ± 10 g, were purchased from the laboratory animal center of Guangxi Medical University, Nanning, Guangxi, China. These animals were kept at 22 °C with free access to food and water during a 12 h light/dark cycle (animal permit number: SCXK (Gui) 2014-0002). After 14 days of adaptation, 3 rats were randomly assigned to be fed a normal diet (ND) as a normal control group, and the other 12 rats were fed a high-fat diet (HFD) as a model control group. The formulas of HFD and ND feed are shown in Appendix A. After 14 days of feeding, all rats fasted for 12 h and blood samples were collected from the abdominal aorta and portal vein. The TC, TG, and LDL-C contents were measured colorimetrically using commercial kits. As indicated in Appendix A, by comparing the model group with the normal group, TG and TC or LDL-c in serum were significantly increased (*p* < 0.01), indicating that all the HFD animals met the standard of hyperlipidemia [10].

#### 2.7.2. Preparation of Fecal Inoculums

By stimulating the anus of rats, the feces of HFD-induced and ND-induced rats were collected in sterile EP tubes. Feces and sterile phosphate-buffered saline (PBS, 1 M, pH 7.2) were mixed in a vortex mixer at the ratio of 1:10 (*w*/*v*) and filtered with four layers of gauze. Supernatant was obtained as fecal slurry. The feces of rats in the HFD-induced rats’ group were divided into 4 parts for follow-up experiments.

#### 2.7.3. Fermentation In Vitro

The anaerobic medium was prepared according to the method of Di et al. [21] with minor modifications. In brief, 2 g of peptone, 2 g of malt extract, 0.1 g of NaCl, 0.04 g of K_2_HPO_4_, 2 g of NaHCO_3_, 0.01 g of MgSO_4_·7H_2_O, 0.01 g of CaCl_2_·6H_2_O, 2 g of Tween 80, 0.05 g of hemin, 0.1 mg of vitamin K, 0.5 g of L-cysteine, 0.5 g of bile salt, and 1 mg of resazurin were dissolved in 1.0 L distilled water and then sterilized at 121 °C for 20 min. TPS2 and TPS5 were dissolved into the PBS at a concentration of 5 mg/mL and sterilized at 121 °C for 20 min. Five fermentation groups were set up: (1) fecal of ND-induced rats plus water (ND group), (2) fecal of HFD-induced rats plus water (HFD group), (3) fecal of HFD-induced rats plus TPS2 (TPS2 group), (4) fecal of HFD-induced rats plus TPS5 (TPS5 group), and (5) fecal of HFD-induced rats plus inulin (Inulin group). Each independently repeated fermentation system was 5 mL, including 3.5 mL anaerobic medium, 0.5 mL fecal slurry, and 1 mL polysaccharide solution or water or inulin solution. Fermentation was carried out in vitro on an Anaero Pack system (HopeBio Co., Ltd., Qingdao, China) at 37 °C. The samples at 0, 12, 24, and 48 h fermentation were taken out for further study.

#### 2.7.4. Measurement of pH and Short-Chain Fatty Acids

The pH value was measured by a pH meter (INESA Scientific Instrument Co., Ltd., Shanghai, China).

Digestive samples were centrifuged at 7104× *g* for 15 min at room temperature and the supernatants collected for SCFA analysis. The SCFAs were assayed by the method of Di et al. [21] using an Agilent 6890 N gas chromatography system (Agilent, Santa Clara, CA, USA) equipped with a DP-FFAP column (0.32 mm × 0.25 µm × 30 m) and a flame ionization detector. Nitrogen was set as the carrier gas, with a total flow of 19 mL/min. A 1.0 µL sample was injected into the GC. The experimental conditions were programmed: oven temperature was initially set at 60 °C for 1 min, increased to 120 °C at a rate of 6 °C/min, increased to 220 °C at a rate of 20 °C/min, and then held at 220 °C for 9 min. The temperatures of the injector and detector were identically set to 250 °C.

### 2.8. DNA Extraction and High-Throughput Sequencing

Fecal samples were analyzed by Novogene Co., Ltd. (Beijing, China), for microbiota analysis. DNA was extracted by using the CTAB method. The V3–V4 hypervariable regions of bacterial 16S rDNA were amplified with forward primer 515 F and reverse primer 806 R. The product was purified with GeneJET (Thermo Fisher Scientific Inc., Waltham, MA, USA). For all qualified DNA, the Ion Plus Fragment Library Kit 48 rxns (Thermo Fisher Scientific Inc., Waltham, MA, USA) was used to construct the library, and the Ion S5^TM^XL (Thermo Fisher Scientific Inc. Waltham, MA, USA) was applied to sequencing after Qubit quantification and library detection. The reads were cut in the low-quality part by using Cutadapt (Version 1.9.1, National Bioinformatics Infrastructure Sweden, Uppsala, Sweden, http://cutadapt.readthedocs.io/en/stable/, accessed on 1 December 2021) and the chimeric sequence was removed to obtain the final effective data (clean reads). Operational taxonomic unit (OTU) with 97% sequences similarity was generated using Uparse (Version 7.0.1001, Robert C. Edgar, Tiburon, CA, USA, http://www.drive5.com/uparse/, accessed on 5 December 2021). Species annotation analysis (with a set threshold of 0.8 to 1) was performed using the Mothur method with the SSUrRNA database of SILVA (Version 132, Max Planck Institute for Marine Microbiology and Jacobs University, Bremen, Germany, http://www.arb-silva.de/, accessed on 13 December 2021). Observed-otus, Shannon, and the construction of the unweighted pair-group method with arithmetic mean (UPGMA) sample clustering trees were calculated using Qiime software (Version 1.9.1, developed primarily in Caporaso labs, Northern Arizona University, Flagstaff, AZ, USA). Alpha diversity index intergroup variance analysis was performed using R software (Version 2.15.3, R Foundation, Vienna, Austria), and dilution curves and principal coordinate analysis (PCoA) curves were plotted. Linear discriminant analysis effect size (LEFse), which indicates the specific bacteria with differences among groups, was performed using Galaxy (Version 1.0, Harvard School of Public Health, Cambridge, MA, USA, http://huttenhower.sph.harvard.edu/galaxy/, accessed on 18 December 2021).

### 2.9. Statistical Analysis

Data were analyzed by one-way ANOVA using SPSS version 19.0 (SPSS Inc., Chicago, IL, USA) and Dunnett’s *t*-test was conducted for means comparison at *p* values of less than 0.05 or 0.01. All data were expressed as the mean ± standard deviation of the three measurements.

## 3. Results and Discussion

### 3.1. Basic Physicochemical Properties of TPS2 and TPS5

The morphologies of crude TPS, refined TPS, TPS2, and TPS5 are shown in Appendix A. The yields of crude TPS and refined TPS were 10.53% and 4.99%, respectively. The yields of refined TPSs from green tea by different methods ranged from 3.98% to 4.52%, which was slightly lower than our result [22]. This may be due to the fact that fermentation increased the content of TPS [23]. As shown in Figure 1a,b, TPS2 and TPS5 were the main fractions of refined Liupao TPS. Thus, they were chosen for the study. The yields of TPS2 and TPS5 were 24.5% and 29.0%, respectively, corresponding to the refined TPS. The four purified polysaccharide fractions isolated from Fuzhuan tea accounted for 3.0%, 10.4%, 37.7%, and 7.1% of the refined TPS, which was similar to our result [6].

As shown in Table 1, TPS2 contained only carbohydrates (total sugar and uronic acid), while TPS5 was composed of carbohydrates, polyphenols, and proteins. The protein content of purified Liupao TPS reported previously was 0.23–4.64% [11], higher than the content obtained in the current study. This difference could be due to increasing aging time, which enhanced the conjugation between polysaccharides and protein [9]. The molecular weights of TPS2 and TPS5 were 70.591 and 133.867 kDa, respectively, lower than those in purified Fuzhuan TPS at 741 kDa [6]. TPS2 and TPS5 were composed of Man, Rha, GlcA, Glc, Gal, and Ara in the molar ratios of 0.12:0.69:0.20:0.088:1.60:0.37 and 0.090:0.36:0.42:0.07:1.10:0.16, respectively. Gal is the main component, which is consistent with the previous study [11].

As shown in Figure 1c, the strong absorption peak of TPS2 and TPS5 near 3400 cm^−1^ could be attributed to the stretching vibration peak of hydrogen-oxygen bonds, indicating the existence of the hydroxyl group. The peak near 2940 cm^−1^ was the stretching vibration peak of the carbon-hydrogen bond. The strong absorption peaks near 1600, 1410, and 1330 cm^−1^ were attributed to the existence of carboxyl and carbonyl, indicating the presence of uronic acid in TPS2 and TPS5. Three obvious absorption peaks observed in the wavelength range of 1200–1000 cm^−1^ showed the pyran ring in TPS2 and TPS5.

As shown in Figure 1d, TPS5 had strong absorption peaks near the 280 nm wavelength, indicating the presence of protein and nucleic acid. There was a small absorption peak of TPS2 near the 280 nm, suggesting the presence of a small amount of binding phenols, since neither the protein content nor the free phenol content of TPS2 was detected, and the infrared spectra of TPS2 and TPS5 are similar.

### 3.2. In Vitro Simulated Saliva–Gastrointestinal Digestion

#### 3.2.1. Changes in Molecular Weights and Concentration of Reducing Sugars

As shown in Figure 2a–c, no obvious changes were observed in the retention times and response values of TPS2 after saliva–gastrointestinal digestion. The molecular weight and reducing sugar of TPS2 also did not change (Table 2), indicating indigestibility. The Fuzhuan TPS could not be decomposed by digestive juice [20], a finding consistent with the result of the present study. The peak of TPS5 shifted slightly to the right, and the retention time enhanced after intestinal digestion (Figure 2f). In particular, the molecular weight of TPS5 significantly decreased from 133.867 ± 1.003 kDa to 122.252 ± 1.180 kDa after 6 h (*p* ˂ 0.05), indicating that TPS5 was digested. The degradation of polysaccharides, with minimal changes in molecular weight, was not observed clearly in the HPGPC chromatograms. Therefore, the content of reducing sugar to depict degradation, caused by the disruption of aggregates or the chain scission of glycosidic bonds, should be further measured. As shown in Table 2, a significant increase in the reducing sugar content of TPS5 with intestinal digestion indicated that the glycosidic bonds continually broke down into reducing ends. The long incubation time could be due to the degradation of polysaccharides in the intestinal juice.

#### 3.2.2. Changes in Free Monosaccharide

The results showed that TPS2 and TPS5 did not produce free monosaccharides during the digestion process (Appendix A). Usually, the reduction in molecular weight due to the breakdown of glycosidic bonds is accompanied by the generation of free monosaccharides. However, no free monosaccharide in TPS5 could be detected after simulated digestion. This result is consistent with the work of Liu [24], who reported that the polysaccharide from *Wolfberry* exhibited a slight change, but no free monosaccharide was generated throughout the simulated digestion. Hence, the glycosidic bonds of TPS5 were probably broken down by the simulated intestinal condition, and the degradation of TPS5 did not produce free monosaccharides.

#### 3.2.3. Changes in Tea Polyphenols in TPS5

As shown in Appendix A, the polyphenol content of TPS5 increased significantly (*p* < 0.05) from 17.23 to 208.86 µg/mL after digestion, which is in line with the work reported by Wu et al. [18], showing that the polyphenol content of Liupao tea increased after gastrointestinal digestion. This suggests that the phenolic polymers of TPS5 were degraded in the gastrointestinal tract, resulting in more phenolic hydroxyl groups. A sample matrix containing sugars and a small amount of protein could potentially interact with polyphenols physically confined in the sample matrix. Ortega [25] stated that phenolic acids could be liberated from soluble esters connected to fiber, primarily yielding free phenolic acids.

#### 3.2.4. Changes in Antioxidant Activity

Results of the antioxidant activities are expressed as vitamin C equivalent antioxidant capacity (VCEAC), as shown in Figure 3. The ABTS scavenging capacities of TPS2 and TPS5 were 50.91% ± 1.71% and 50.94% ± 00.21% mg/mL VCEAC, respectively, while their DPPH scavenging capacities were 2.89% ± 0.10% and 8.76% ± 0.38% mg/mL VCEAC, respectively. In terms of DPPH assay, the trend of antioxidant activity was different from ABTS assay. This difference could be ascribed to ABTS, which is usually used as a tool to estimate the total antioxidant activity. DPPH is usually used as an indicator for assessing free radical scavenging activities [26]. Very few studies have reported the antioxidant activity of polysaccharides after the digestion process. As shown in Figure 3a, the ABTS capacities of TPS2 and TPS5 gradually declined in the oral, gastric, and intestinal phases. Figure 3b shows that the DPPH capacities of TPS2 and TPS5 decreased during oral digestion, increased after gastric digestion, and then decreased after intestinal digestion.

TPS5 consistently had a higher antioxidant capacity than TPS2. The antioxidant capacity of polysaccharides is influenced by the component of TPS. For example, the higher uronic acid in Puerh TPS could lead to higher antioxidant capacity, possibly because the carboxyl group on uronic acid could play the role of hydrogen donor and electron transfer agent [5]. The effects of the protein content of Puerh TPS on the improvement of bioactivities appeared to be significant [5]. The antioxidant activity of Qingzhuan TPS was related to its polyphenols [8]. Given the large amount of polyphenols released by TPS5 after digestion (Appendix A), it is inferred that the antioxidant activity of TPS5 is affected by the content of polyphenols, possibly by the content of uronic acid or protein. The structure of polysaccharides can also influence their antioxidant activity. For example, the *Epimedium acuminatum* Franch polysaccharide EAP40-1 and EAP60-1 with its higher molecular weight showed higher antioxidant capacity than EAP80-2 [27]. The higher the branching degree of the sugar chain of TPS, the better the space conditions for contacting with free radicals, resulting in better antioxidant capacity [28]. Thus, it can be inferred that the large molecular weight also has an effect on the antioxidant activity of TPS5, and TPS5 may have a branching structure. The antioxidant activities of TPS2 and TPS5 were altered during the simulated digestion phase, probably due to changes in hydroxyl and glucoronide content [26]. The protein in TPS5 may also affect the antioxidant activity in the digestive process probably due to the exposure of protein to more hydrophobic amino acid side chains during simulated gastric digestion by pepsin. This makes the peptides more accessible to the radicals and allows the peptides to be easily trapped by the radical. During intestinal digestion, proteins are more thoroughly hydrolyzed into hydrophilic oligopeptides and amino acids, and the increased polarity makes trapping free radicals difficult [29]. Furthermore, TPS5 was digested into smaller molecular-weight polysaccharides and released large amounts of polyphenols during the intestinal phase, which may also affect the activity. Overall, the results indicate that polysaccharide conjugates containing polyphenols and proteins have better bioactivity.

### 3.3. Ability to Bind Bile Acids

Table 3 shows the bile acid binding capacities of TPS2 and TPS5, which were 42.79% ± 1.56% and 33.78% ± 0.45%, respectively, lower than the results of a previous study [11]. A possible reason was that a higher extraction temperature was used in the present study. The binding capacity of bile acids and polysaccharides is related hydrophobic interactions, electrostatic interactions, and hydrodynamic limitations, together with monosaccharide composition, macromolecular characteristics, viscosity, and surrounding bile acids of polysaccharides. The bile acid binding capacity of Liupao TPS was not associated with polyphenols, as was found in the study by Qin et al. [11]. This could be a result of the high viscosity due to the higher molecular weight of TPS5, which has a hydrodynamic limiting effect on bile acid binding, while the lower molecular weight of TPS2 is more prone to produce active polymeric structures [30]. Similarly, the β-glucan having lower molecular weight had a better bile acid binding activity [31]. A kelp polysaccharide with high Gal had stronger bile acid binding capability, probably related to its highly branched structure and Gal on the side chain [32]. Therefore, the better bile acid binding capacity of TPS2 may be explained by lower molecular weight and higher Gal. In addition, the type of glycosidic bond of the polysaccharide can also influence its activity; for example, a fraction consisting of 1 → 3 linked glucans isolated from pumpkin polysaccharide was more active than the fraction consisting of 1 → 4 bond [28].

### 3.4. In Vitro Fermentation

#### 3.4.1. Changes in pH and SCFAs

As shown in Appendix A, the initial pH was similar for all groups at around 8.0. During digestion, the pH of the polysaccharide was lower than that of the HFD but higher than that of the positive group, consistent with the fermentation of Fuzhuan TPS [6,20]. After 48 h, the pH of TPS2 and TPS5 significantly decreased to 6.14 and 6.36 (*p* < 0.05), respectively. TPS2 exhibited stronger capability to reduce pH. This can be explained by the lower molecular weight of TPS2, which indicates it was consumed faster by intestinal microorganisms, producing lower pH [33]. Therefore, the effect of TPS2 on improving the intestinal microenvironment was better.

As shown in Table 4, the concentration of total SCFAs increased significantly in all groups after 48 h. The total SCFAs of inulin, TPS2, and TPS5 groups were significantly higher than those of the HFD and ND groups (*p* < 0.05), suggesting that polysaccharides and inulin could modulate the intestinal microenvironment to produce SCFAs. During the fermentation of TPS2 and TPS5, the content of acetic acid was highest, followed by propionic acid, which is consistent with the green TPS fermentation [34]. After 48 h, the acetic acid and propionic acid in the TPS2 group were significantly higher than those in the TPS5 group (*p* < 0.05). The possible reason is that the fermentation of the high proportion of Gal in TPS2 led to the increase in acetic acid, while that of Glc, Ara, and Man led to the increase in propionic acid [35]. The acetic acid content of TPS5 reached a maximum at 24 h and decreased at 48 h, which is consistent with the fermentation of red kidney bean polysaccharide, and it might be due to the change in the relative abundance of *Lactobacillus* in TPS5 [36]. The increase in population of *Lactobacillus* spp. could be responsible for the higher production of acetic acid [36]. Acetic acid is the key factor to inhibit intestinal pathogenic bacteria, and propionic acid could regulate immune cells and increase the production of antibiotic features [35]. The butyric acid of TPS5 was significantly higher than that of TPS2 (*p* < 0.05). TPS5 has higher abundance of *Roseburia*, which is associated with the production of butyric acid [34]. Butyric acid is the energy substrate of colon cells, and enhances the function of intestinal epithelial barrier [20]. In addition, a low proportion of acetic acid to propionic acid is assumed to prevent the liver biosynthesis of cholesterol and fatty acids [37]. After 48 h, the ratio of acetic acid to propionic acid in TPS5 (3.05 ± 0.19) was lower than that in TPS2 (4.26 ± 1.44), indicating that TPS5 may have a better ability to inhibit cholesterol synthesis. Overall, the change in SCFAs content during fermentation basically conforms to the change in pH. Based on the different abilities to promote SCFAs production, TPS2 may regulate host metabolism better than TPS5. Previous studies have shown that fucogalactan sulfate extracted from *Laminaria japonica* had a better effect on promoting the production of SCFAs, which has highly branched sugar residue such as (1 → 2, 3, 4) linked β-D-ManpA and sulfate ester groups, suggesting that TPS2 may have a similar structure [38].

#### 3.4.2. Effects on Gut Microbiota

The bacterial lineages across fecal samples after in vitro fermentation were characterized to evaluate the effect of TPS2 and TPS5 on gut microbiota. A total of 1,256,458 raw reads were obtained for forward and reverse directions after sequencing; 1,193,282 clean reads were retained after the preliminary quality filtering (Appendix A). Rarefaction analysis revealed that the curves did not reach a plateau (Appendix A); however, the Shannon indices of all samples remained steady (Appendix A). This suggests that the sequencing process captured most bacterial species and could exactly characterize the microbial community of the samples.

As shown in Appendix A, the first two principal coordinates interpreted 48.71% of the total intersample variance (PC1: 33.3% and PC2: 15.41%). The structure of intestinal flora in the TPS2 and TPS5 groups was similar but different from other groups. The same results can be observed in Appendix A.

Figure 4a shows the taxonomic composition distribution of samples at the phylum level. The dominant bacterial phyla in ND groups were *Proteobacteria*, *Bacteroidetes*, *Firmicutes*, and *Actinobacteria*, consistent with the four phyla with the highest abundance in human fecal fermentation in vitro for 48 h [39], indicating in vitro fermentation of rat feces can partly reflect the fermentation characteristics of human feces. The proportion of *Firmicutes* to *Bacteroidetes* (F/B) is considered as a biological indicator of obesity, and it is higher in adipose persons having a high-calorie diet [20]. The TPS from green tea and Fuzhuan tea could decrease the F/B ratio [20,34], and a similar effect has been observed in Liupao tea extracts [40]. As shown in Appendix A, comparing to the HFD group, fermentation of TPS2 and TPS5 led to a decrease in the F/B ratio to 0.33 and 0.36, respectively, suggesting that TPS2 may have a better effect on weight loss. By contrast, inulin resulted in an increase in F/B ratio, which might be attributed to the lower pH after fermentation. A number of *Firmicutes* species presents the feature of low pH tolerance, but many *Bacteroidetes* cannot survive under a low pH environment [21]. The family-level classification of microbial communities is presented in Figure 4b. The *Pseudomonadaceae* in TPS2 and TPS5 groups was significantly enriched (*p* ˂ 0.05) compared to that in the ND, HFD, and inulin groups (Appendix A). Some species of *Pseudomonadaceae* are pathogenic to humans, while some of them exhibit virulence against cervical cancer cells and inhibitory activity against harmful bacteria [41]. In comparison to the HFD group, the TPS2 groups showed decreased abundances of *Enterococcaceae* and *Lachnospiraceae* to 8.9% and 0.5%, respectively, while the TPS5 group exhibited 12.0% and 1.0% decrease, respectively (Appendix A). The changes in intestinal flora in patients with nonalcoholic fatty liver included an increase in *Enterococcaceae* and *Lachnospiraceae* [21]. This finding suggests that TPS2 may play a better role in protecting the liver. At the genus level (Figure 4c), the abundance of *Bacteroids* in the TPS2 and TPS5 groups increased significantly (*p* ˂ 0.05) to 28.0% and 33.6%, respectively, whereas that of *Proteus* decreased significantly (*p* ˂ 0.05) to 0.6% and 0.9%, respectively, compared to the HFD group (Appendix A). Liupao tea polyphenols could enhance *Bacteroids* in the intestinal flora of high-fat mice [40], which could possibly be the reason for the higher abundance of *Bacteroids* in the TPS5 group. The higher relative abundance of *Proteus* was related to frog diarrhea [42], suggesting a better therapeutic effect of TSP2 for diarrhea. TPS2 and TPS5 led to the reduction in *Lactobacillus* (Appendix A). A similar result was obtained in the fermentation of Liupao tea extract [40].

A popular method to determine the key gut microbiota is the linear discriminant analysis effect size (LEfSe). In Figure 4d, two families (unidentified *Bacteroidales* and *Moraxellaceae*) of TPS2 are higher than those of TPS5, while two families (*Erysipelotrichaceae* and *Ruminococcaceae*) of TPS5 are higher than those of TPS2. Therefore, the microbial composition of TPS2 and TPS5 is similar in this study, consistent with the results of PCoA. As shown in Figure 4e, 7 OTUs in the TPS5 group, 9 OTUs in the TPS2 group, 45 OTUs in the ND group, 13 OTUs in the inulin group, and 11 OTUs in the HFD group were significantly higher than those in the other three groups. For the TPS5 group, family *Erysipelotrichia*, *Bacteroides salyersiae*, *Bacteroides nordii*, and genus *Roseburia* were increased. *Erysipelotrichaceae* was enriched after fermentation of fructooligosaccharide with human intestinal flora [34]. This finding may indicate that TPS5 plays a prebiotic role as fructooligosaccharide. *Erysipelotrichaceae* has the ability to metabolize plant polysaccharides, and their abundance was positively correlated with the level of SCFAs [43]. *Bacteroides salyersiae* is able to break down mannose [44]. Tea polyphenols could increase the abundance of *Roseburia* in mice with intestinal disorders, and *Roseburia* is part of commensal bacteria producing SCFAs [40]. For the TPS2 group, the four microorganisms with the highest scores belong to *Pseudomonadales*. *Bacteroides dorei*, *Bacteroides vulgatus*, and *Bacteroides thetaiotaomicron* belong to *Bacteroides*. The relative abundance of *Dysgonomonas* was also increased. *Bacteroides dorei* could ferment dietary fiber polysaccharides, and it was associated with reduced inflammation [20]. *Bacteroides thetaiotaomicron* produces multiple glycoside hydrolases to metabolize and uses complex dietary glycans [21]. *Dysgonomonas* hydrolyzes starch and produces SCFAs from polysaccharide building blocks, such as arabinose, cellobiose, rhamnose, and xylose [45]. Altogether, fermentation of TPS2 and TPS5 changed the microbial structure and increased the relative abundances of some bacteria that could depolymerize and metabolize polysaccharide. The active regions in TPS that regulate intestinal flora were the arabinogalactan and rhamnose galacturonic acid structure [28]. For example, polysaccharides from citrus segment membranes with high rhamnogalacturonan content were beneficial to the ecology of intestinal microorganisms [46]. Different intestinal bacteria have different preferences for different types of polysaccharides [28]. *Bacteroides* may be a common dominant microbiota for the metabolism of TPS2 and TPS5, similar to the previous result of Fuzhuan TPS fermentation in vitro [6]. *Erysipelotrichia* and *Roseburia* are the specific microbiota utilizing TPS5, and *Dysgonomonas* is the specific microbiota utilizing TPS2.

## 4. Conclusions

In this study, TPS2 contained 53.73% ± 1.55% total sugar and 35.18% ± 0.96% uronic acid, while TPS5 contained 51.71% ± 1.16% total sugar, 40.95% ± 3.12% uronic acid, 0.11% ± 0.07% protein, and 0.43% ± 0.03% polyphenol. TPS2 and TPS5 were mainly composed of galactose, with molecular weights of 70.591 and 133.867 kDa, respectively. After digestion was simulated, TPS2 was not degraded. The molecular weight of TPS5 decreased significantly from 133.867 ± 1.003 kDa to 122.252 ± 1.180 kDa (*p* < 0.05) and produced reducing sugar and free polyphenol. The antioxidant capacity of TPS5 was better than that of TPS2 in simulated digestion. The bile acid binding ability of TPS2 was better than that of TPS5. In fermentation in vitro, TPS2 exhibited higher capability to decrease pH and promote production of acetic and propionic acids; meanwhile, TPS5 showed stronger capability to increase butyric acid and reduce the ratio of acetic acid to propionic acid. At the phylum level, TPS2 and TPS5 could increase the abundance of *Bacteroides* and reduce the abundance of *Firmicutes*. TPS2 also had a better effect on reducing the *Firmicutes*-to-*Bacteroidetes* ratio. At the family level, TPS2 was more likely to reduce the abundance of *Enterococcaceae* and *Lachnospiraceae*. At the genus level, TPS5 improved the abundance of *Bacteroids*, while TPS2 diminished the abundance of *Proteus*. The results revealed that the presence of polyphenols and proteins contributed differently to different activities, but TPS2 and TPS5 are potentially beneficial to humans.

## Figures and Tables

**Figure 1 foods-11-02958-f001:**
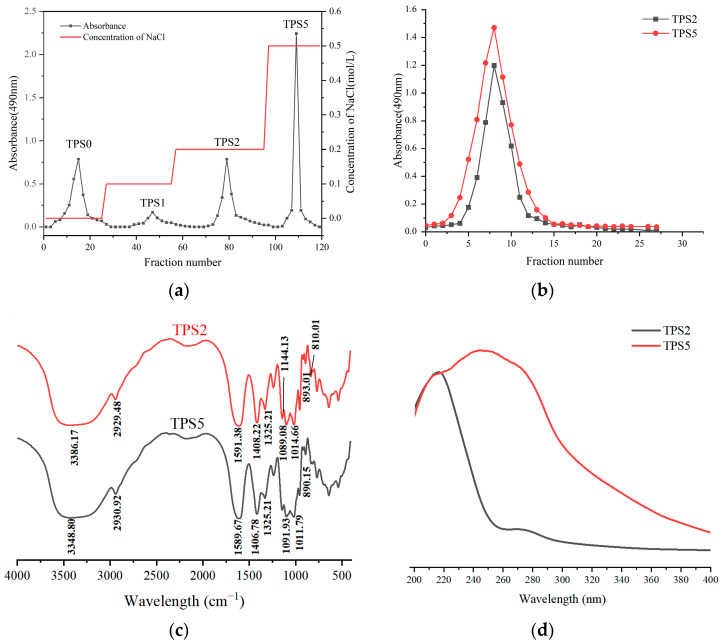
Elution curve of the purification of refined Liupao TPS by DEAE-52 cellulose column (**a**). Elution curve of the enrichment of TPS2 and TPS5 detected by the phenol–sulfuric acid method (**b**). Fourier transform infrared spectrogram of TPS2 and TPS5 (**c**) and ultraviolet-visible spectrogram of TPS2 and TPS5 (**d**).

**Figure 2 foods-11-02958-f002:**
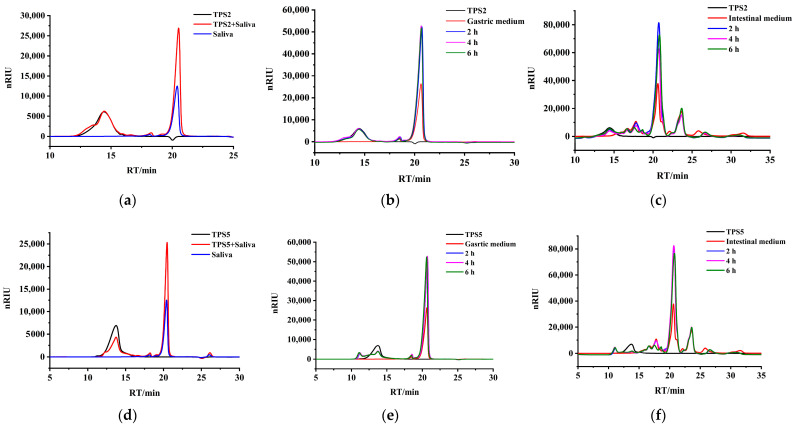
HPGPC chromatograms of TPS2 before and after simulated saliva (**a**), stomach (**b**), and small intestinal (**c**) digestion. HPGPC chromatograms of TPS5 before and after simulated saliva (**d**), stomach (**e**), and small intestinal (**f**) digestion.

**Figure 3 foods-11-02958-f003:**
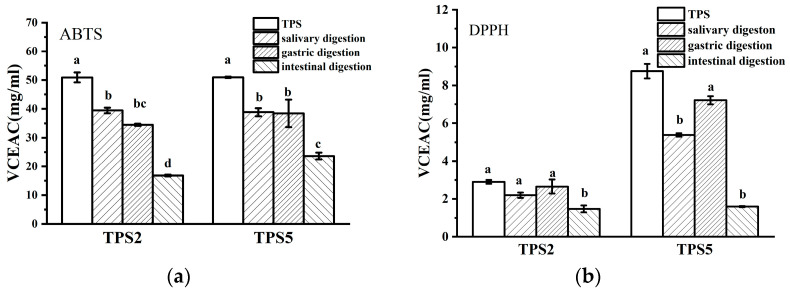
Radical scavenging ability of ABTS (**a**) and DPPH (**b**) of TPS2 and TPS5 in simulated digestion. Different letters indicate significant difference (*p* < 0.05).

**Figure 4 foods-11-02958-f004:**
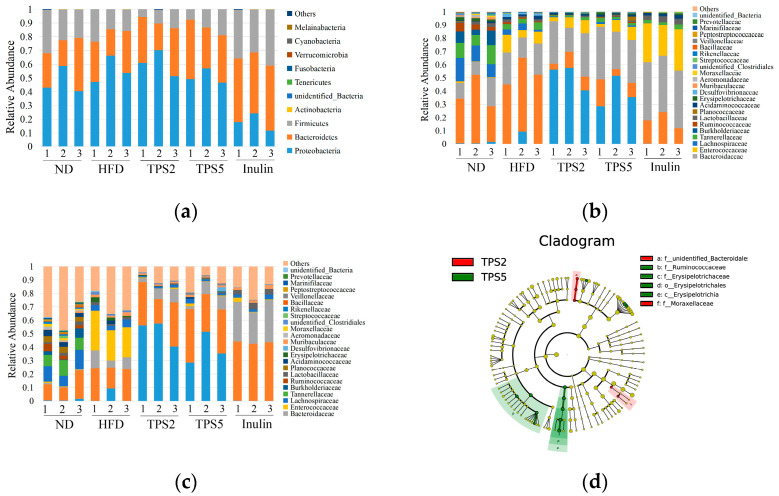
Microbiota composition during in vitro fermentation. Relative abundance of bacterial community at the phylum level (**a**), family level (**b**), and genus level (**c**). Cladogram of LEfSe of 48 h fermentation samples; TPS2 and TPS5 (**d**); ND, HFD, TPS2, TPS5, and inulin (**e**).

**Table 1 foods-11-02958-t001:** Structural characterization of TPS2 and TPS5.

Item	TPS2	TPS5
Total sugar (W%)	53.73 ± 1.55	51.71 ± 1.16
Uronic acid (W%)	35.18 ± 0.96	40.95 ± 3.12
Protein (W%)	ND	0.11 ± 0.07
Polyphenol (W%)	ND	0.43 ± 0.03
Molecular weight (kDa)	70.591	133.867
Monosaccharide composition (mol %)		
Mannose	0.12	0.090
Rhamnose	0.69	0.36
Glucuronic acid	0.20	0.42
Glucose	0.088	0.07
Galactose	1.60	1.10
Arabinose	0.37	0.16

ND means not detected.

**Table 2 foods-11-02958-t002:** Molecular weights and concentrations of reducing sugars of TPS2 and TPS5 during in vitro simulated saliva–gastrointestinal digestion.

Simulated Digestive Juice	Time (h)	Molecular Weight (kDa)		Reducing Sugar (mM)	
		TPS2	TPS5	TPS2	TPS5
Salivary	0	70.591 ± 0.541 ^a^	133.867 ± 1.003 ^a^	0 ^a^	0 ^a^
	1	70.591 ± 0.541 ^a^	133.867 ± 1.003 ^a^	0 ^a^	0 ^a^
Gastric	0	70.591 ± 0.541 ^a^	133.867 ± 1.003 ^a^	0 ^a^	0 ^a^
	2	70.591 ± 0.541 ^a^	133.867 ± 1.003 ^a^	0 ^a^	0 ^a^
	4	70.591 ± 0.541 ^a^	133.867 ± 1.003 ^a^	0 ^a^	0 ^a^
	6	70.591 ± 0.541 ^a^	133.867 ± 1.003 ^a^	0 ^a^	0 ^a^
Intestinal	0	70.591 ± 0.541 ^a^	133.867 ± 1.003 ^a^	0 ^a^	0 ^a^
	2	70.591 ± 0.541 ^a^	124.270 ± 2.120 ^b^	0 ^a^	0.679 ± 0.029 ^b^
	4	70.591 ± 0.541 ^a^	123.256 ± 1.534 ^c^	0 ^a^	0.743 ± 0.009 ^c^
	6	70.591 ± 0.541 ^a^	122.252 ± 1.180 ^d^	0 ^a^	1.028 ± 0.007 ^d^

Different letters in the same column indicate a significant difference (*p* < 0.05).

**Table 3 foods-11-02958-t003:** In vitro bile acid binding ability of TPS2 and TPS5.

Treatment	Bile Acid Concentration in Supernatant (μmol/L)	Percentage Bile Acid Binding (%)	Relative to Cholestyramine (50mg), %
TPS2	7.32 ± 0.20 ^b^	42.79 ± 1.56 ^b^	50.53 ± 1.84 ^b^
TPS5	8.47 ± 0.06 ^a^	33.78 ± 0.45 ^c^	39.89 ± 0.53 ^c^
Cholestyramine	1.95 ± 0.21 ^c^	84.68 ± 1.62 ^a^	100.00 ± 1.92 ^a^

Different letters in the same column indicate significant difference (*p* < 0.05).

**Table 4 foods-11-02958-t004:** Short-chain fatty acids and ratio of acetic acid to propionic acid at different time points of fermentation.

SCFAs (mmol/L)	Groups	Fermentation Time (h)			
		0	12	24	48
Acetic acid	HFD	7.05 ± 0.68 ^a,^ ^A^	3.82 ± 0.22 ^b,^ ^D^	6.75 ± 0.76 ^a,^ ^D^	3.69 ± 0.20 ^b,^ ^D^
	ND	7.18 ± 0.23 ^b, A^	11.76 ± 1.05 ^a, C^	7.07 ± 0.33 ^b, D^	4.57 ± 0.34 ^c, D^
	TPS2	7.10 ± 0.27 ^d, A^	14.05 ± 0.67 ^c, C^	23.25 ± 1.44 ^b, B^	26.63 ± 1.38 ^a, B^
	TPS5	7.54 ± 0.07 ^d, A^	26.00 ± 0.50 ^b, A^	27.52 ± 0.33 ^a, A^	12.29 ± 0.93 ^c, C^
	Inulin	7.89 ± 0.28 ^c, A^	18.77 ± 2.35 ^b, B^	16.99 ± 0.44 ^b, C^	52.44 ± 1.64 ^a, A^
Propionic acid	HFD	0.62 ± 0.08 ^d, A^	1.36 ± 0.07 ^c, C^	5.19 ± 0.27 ^b, A^	8.67 ± 0.22 ^a, B^
	ND	0.72 ± 0.01 ^d, A^	5.44 ± 0.23 ^a, A^	3.24 ± 0.16 ^b, C^	2.58 ± 0.23 ^c, C^
	TPS2	0.69 ± 0.20 ^c, A^	1.49 ± 0.01 ^c, C^	3.94 ± 0.15 ^b, B^	6.64 ± 1.76 ^a, B^
	TPS5	0.56 ± 0.05 ^c, A^	4.08 ± 0.03 ^a, B^	3.65 ± 0.02 ^b, BC^	4.04 ± 0.29 ^a, C^
	Inulin	0.62 ± 0.02 ^c, A^	0.62 ± 0.07 ^c, D^	1.49 ± 0.30 ^b, D^	31.13 ± 0.56 ^a, A^
Butyric acid	HFD	0.36 ± 0.06 ^a, A^	0.45 ± 0.07 ^a, B^	0.37 ± 0.01 ^a, C^	0.31 ± 0.08 ^a, E^
	ND	0.29 ± 0.04 ^c, A^	2.45 ± 0.11 ^b, A^	2.62 ± 0.06 ^b, A^	2.84 ± 0.06 ^a, A^
	TPS2	0.38 ± 0.03 ^b, A^	0.61 ± 0.15 ^ab, B^	0.41 ± 0.08 ^b, C^	0.71 ± 0.07 ^a, D^
	TPS5	0.38 ± 0.13 ^c, A^	0.72 ± 0.13 ^b, B^	0.90 ± 0.06 ^b, B^	1.64 ± 0.09 ^a, B^
	Inulin	0.29 ± 0.04 ^c, A^	0.61 ± 0.01 ^b, B^	0.26 ± 0.04 ^c, C^	1.16 ± 0.09 ^a, C^
Total	HFD	8.03 ± 0.62 ^c, A^	5.63 ± 0.23 ^b, D^	12.31 ± 0.51 ^a, D^	12.67 ± 0.21 ^a, D^
	ND	8.20 ± 0.24 ^c, A^	19.65 ± 1.38 ^a, B^	12.93 ± 0.30 ^b, D^	10.00 ± 0.34 ^c, D^
	TPS2	8.17 ± 0.39 ^d, A^	16.15 ± 0.73 ^c, C^	27.60 ± 1.61 ^b, B^	33.97 ± 1.83 ^a, B^
	TPS5	8.48 ± 0.07 ^d, A^	30.80 ± 0.60 ^b, A^	32.07 ± 0.30 ^a, A^	17.97 ± 1.17 ^c, C^
	Inulin	8.80 ± 0.27 ^c, A^	20.00 ± 2.28 ^b, B^	18.74 ± 0.63 ^b, C^	84.72 ± 1.04 ^a, A^
Acetic acid/Propionic acid	HFD	11.41 ± 1.83 ^a, A^	2.82 ± 0.26 ^b, B^	1.31 ± 0.21 ^b, C^	0.43 ± 0.03 ^b, D^
	ND	9.92 ± 0.36 ^a, A^	2.16 ± 0.11 ^b, B^	2.19 ± 0.19 ^b, C^	1.78 ± 0.27 ^b, BC^
	TPS2	10.97 ± 3.29 ^a, A^	9.43 ± 0.40 ^ab, B^	5.90 ± 0.19 ^bc, B^	4.26 ± 1.44 ^c, A^
	TPS5	13.44 ± 1.19 ^a, A^	6.38 ± 0.12 ^b, B^	7.54 ± 0.06 ^b, B^	3.05 ± 0.19 ^c, AB^
	Inulin	12.77 ± 0.13 ^b, A^	30.78 ± 7.03 ^a, A^	11.75 ± 2.46 ^b, A^	1.69 ± 0.08 ^c, BC^

Different minuscules show significant differences at different time (*p* < 0.05) in the same group, while capitals represent significant differences among different groups (*p* < 0.05) at the same time point.

## Data Availability

The sequencing data have been deposited at the National Center for Biotechnology Information (https://www.ncbi.nlm.nih.gov/, accessed on 3 April 2022) with the accession number PRJNA822778.

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
