# Peer review of "Two Polysaccharides from Liupao Tea Exert Beneficial Effects in Simulated Digestion and Fermentation Model In Vitro"

_foods, 2022, doi:10.3390/foods11192958_

Round 1

Reviewer 1 Report

foods-1901594-peer-review-v1

This is interesting work, complex evaluation of the bioactive compound, from isolation, purification, characterization and evaluation of their beneficial roles.

In my opinion paper deserve to be recommended for publication, however, an revision, including some adjustments, correction, better presentation and upgrade need to be taken into consideration by the authors.

Some specific comments

Ln17: Please, remove interval before ",".

Ln29: Maybe will be appropriate to mention in what province is located Liupao town.

Ln31: No need for italics for "var." and second sinensis do not need to be with capital.

Ln33: Do you know what exact species of Aspergillus are involve in the fermentation process? This is important point, since some of the Aspergillus are well known as mycotoxin producers, and other are human and other animals’ pathogens.  

Ln40. Maybe is a better way to say "active substances "?

Ln98: Please, correct to (Sigma-Aldrich, ...provide cite, state, ... USA).

Ln110: What was extracted from Liupao tea? Please, state that TPS2 and TPS5 were extracted from Liupao tea...

Ln111: specify role of the filter in previous sentence.

Ln129: Really need two references for a same method?

Ln146: Maybe present values in milligrams, in order for better unification. Previous values were presented in milligrams.

Ln225: Provide information about the applied kit.

Ln232: Can authors provide copy of this certificate? It is a bit strange to have Ethical comity from 2014, applying for the work in 2022.

Ln236: Provide address for this medical university, city, province, country.

Ln252: italics for "in vitro"

Ln258: how the sterilization been performed?

Ln270: Centrifugation need to be as "xg"

Ln305: If I am not wrong, recommendations form the journal are that Results and Discussion need to be separated. Please, follow instructions for authors.

Please, try do not start sections with table or figure, normally each section needs to start with a short text.

Figure 4 is very difficult to read. Maybe authors can increase size of the textual part?

Discussion needs a bit more attention, specially first yew experiments described in the section, where practically only fact of the experimental parts are reported and no discussion provided.

Reviewer 2 Report

The work reported by Qui et al. has a very good and original results. Authors showed beneficial effects of purified polysaccharides in simulated digestion and fermentation model in vitro.

I recommend the publication of this article after making minor revisions/

1. In Abstract authors must include results of sugar composition analysis.

2. In The introduction authors must add chinese national statistics about Liupao tea production.

3. How about the yield of crude TPS production and for TPS2 and TPS5, results must be discussed and compared with recent literature.

4. Authors report that TPS5 is composed of sugar, proteins and polyphenol, so the reported antioxidant activity is affected for which component?

5. TPS2 is exempt of proteins but in Fig 1d there is a small peak for TPS2, please clarify?

6. In results and discussion section authors must add some discussion about the structure function relationship.
